

# Individual and demographic responses of the palm *Brahea aculeata* to browsing and leaf harvesting in a tropical dry forest of Northwestern Mexico

Franceli Macedo-Santana[1], Christa Horn[2], Tamara Ticktin[3], María Teresa Pulido Silva[4], Bryan A. Endress[5] and Leonel Lopez-Toledo[1]

[1] Instituto de Investigaciones sobre los Recursos Naturales, Universidad Michoacana de San Nicolás de Hidalgo, Morelia, Michoacan, Mexico

[2] San Diego Zoo Wildlife Alliance, Escondido, CA, United States of America

[3] Department of Botany, University of Hawaii at Manoa, Honolulu, Hawa'i, United States of America

[4] Centro de Investigaciones Biológicas, Universidad Autónoma del Estado de Hidalgo, Pachuca, Hidalgo, Mexico

[5] Eastern Oregon Agriculture and Natural Resource Program, Oregon State University, La Grande, OR, United States of America

Corresponding authors
Bryan A. Endress,
Bryan.Endress@oregonstate.edu
Leonel Lopez-Toledo,
leonellopeztoledo@gmail.com,
llopezt@umich.mx

## ABSTRACT

**Background**. The leaves of many palm species represent important non-timber forest products (NTFPs), which may be intensively harvested by local people in many tropical areas. Additionally, in some regions livestock graze in natural forests, and they may browse on palm leaves, especially during the dry season. Thus, harvesting and browsing can result in the loss of leaf area of individual palms, which may alter functional traits of individuals and change demographic patterns of populations. Currently, there are few studies that analyze the effects of multiple disturbances on these traits. The goals of this study were to evaluate the effects of browsing, leaf harvesting and the interaction between these two factors on individual traits and demographic patterns of the *Brahea aculeata* palm in northwestern Mexico.

**Methods**. A browsing and leaf harvesting experiment was conducted on natural populations of the species. Individuals were subjected to different harvesting intensities and the presence or absence of cattle. Annual censuses were conducted from 2011 to 2014, and individual traits (leaf length, petiole length, and leaf production) and vital rates were monitored.

**Results**. At the individual level, the analyzed traits mostly increased as function of leaf harvest and browing, especially during the first two years. Palms experiencing leaf harvesting and browsing had 1.5 to 6.0 times higher levels of leaf production than control palms, especially juveniles and small adults. At the demographic level, the effects of browsing and leaf harvest were low or null, since survival was not affected by them. Browsing positively affected the growth of *B. aculeata* individuals in the first 2 years, while leaf harvesting had a negative effect in year three. There was a positive relationship between the probability of reproduction and leaf harvest; however, high leaf harvest resulted in two to three times fewer fruits produced. After 3 years of experimental management, multiple of the analyzed attributes decreased, suggesting that *B. aculeata* changed patterns of resource allocation. Based on our results, *B. aculeata* can be considered a species that tolerates high levels of defoliation and browsing for 2

years, but not likely longer. This study contributes basic ecological information useful for the conservation and management of *B. aculeata*, but overall it also highlights that different anthropogenic activities may act as drivers affecting the functional response and demography of NTFP species and they should be considered for the long-term integral management of these species.

## INTRODUCTION

The harvest of non-timber forest products (NTFPs) represents an important source of income and contributes to the welfare of many rural communities across the globe (*Krishnakumar, Fox & Anitha, 2012*; *Shujaul Mulk Khan et al., 2020*; *Shackleton et al., 2024*). Intensive use and overexploitation of NTFPs, however, may negatively affect individuals and populations (*Endress, Gorchov & Berry, 2006*; *Lopez-Toledo et al., 2012*). Nevertheless, with appropriate management practices, the harvest of NTFPs may also contribute to the conservation of natural resources and biodiversity (*Gaoue et al., 2016*; *Rodrigues de Mello et al., 2023*).

The study of the responses of functional individual traits to stress factors, such as leaf area loss (caused by leaf harvesting or browsing), can provide insight regarding the anatomical and physiological responses of plants and how this relates to changes at the population level (*Briske & Richards, 1995*; *Poorter, 1999*; *Violle et al., 2007*). Vital rates (reproduction, growth and survival) are also important for exploring response to disturbances such as management practices and for identifying the sustainable use of NTFPs (*Zuidema, De Kroon & Werger, 2007*; *Martínez-Ballesté & Martorell, 2015*; *Ohse et al., 2023*). Thus, studying the effects of different harvest and management practices on both individual and demographic traits of harvested populations may help to identify optimal management strategies (*Anten, Martinez-Ramos & Ackerly, 2003*; *Hernández-Barrios et al., 2012*; *Gaoue et al., 2016*; *Ticktin et al., 2023*).

Leaves are important organs for essential processes, such as capturing light energy, carbohydrate production and water conservation (*Wright et al., 2004*). Thus, the loss of leaf area can affect plants' essential functions, such as growth, reproduction and/or individuals' survival.

Plants usually undergo multiple leaf area loss events caused by reoccurring biotic and physical damage (herbivory, fallen branches; (*Martínez-Ramos & Álvarez-Buylla, 1995*; *Cepeda-Cornejo & Dirzo, 2010*). For plants used as NTFPs, leaf harvesting may represent a large increase in the amount and frequency of leaf area loss. This reduction in leaf material may alter the allocation of resources to different plant functions, such as reproduction, growth and maintenance. Plants may compensate for the reduction in leaf material by shifting resources from reproduction to leaf production and/or mobilizing stored reserves to produce new leaves (*McNaughton, 1983*; *Belsky et al., 1993*; *Anten, Martinez-Ramos &*

*Ackerly, 2003*; *Lopez-Toledo et al., 2012*; *Sun, Sharifi & Rundel, 2022*). In scenarios with higher frequency and/or higher intensities of leaf area loss, which can occur during the harvest of some NTFPs, plants' capacity to compensate can be reduced, however, as stored reserves are depleted, potentially resulting in growth reduction and increased mortality (*Farrington et al., 2008*; *Martínez-Ramos, Anten & Ackerly, 2009*; *Lopez-Toledo et al., 2012*; *Ward, Jones & Barsky, 2022*).

Many palm species are culturally and/or economically important NTFPs-providing species. The leaves of several palm species are used for many products, including roof thatch and handicrafts, and represent an important source of income for local economies especially in rural areas (*Svenning & Macia, 2002*; *Coronel & Pulido, 2010*; *Briseño Tellez et al., 2023*; *Sander, Pulido-Silva & Da Silva, 2023*). Multiple studies have been conducted on palms used as NTFPs to analyze the effects of leaf harvesting on leaf production and the vital rates of individuals (*Endress, Gorchov & Berry, 2006*; *Hernández-Barrios et al., 2012*; *Lopez-Toledo et al., 2012*; *Martínez-Ramos, Anten & Ackerly, 2009*; *Mandle & Ticktin, 2012*; *Martínez-Ballesté & Martorell, 2015*). Some studies reported no effect of harvesting on individual palm's vital rates (growth, reproduction and survival) when the harvest is restricted to a few defoliation events, even with complete defoliation (*Oyama & Mendoza, 1990*; *Pulido & Coronel-Ortega, 2015*). In some cases, defoliated plants even show elevated levels of growth and reproduction because of overcompensation (*Anten, Martinez-Ramos & Ackerly, 2003*; *Martínez-Ramos, Anten & Ackerly, 2009*; *Gaoue et al., 2016*). Other studies, however, indicated that repeated defoliation over multiple years negatively affects leaf production and vital rates, and these negative effects intensify with increased harvest intensity or frequency (*Martínez-Ramos, Anten & Ackerly, 2009*; *Hernández-Barrios et al., 2012*; *Lopez-Toledo et al., 2012*).

In addition, in many tropical regions, cattle ranching is a common land use in forests, grasslands and other natural areas (*Herrera, 1995*; *Meghan, Graydon & Cushman, 2013*), and livestock may also browse or/graze on species that are also NTFPs. It has also been shown that cattle may modify soil compaction and other physical and chemical properties (*Fleischner, 1994*), and, in the long term, these effects may become evident in the structure (*Stern, Quesada & Stoner, 2002*) and in the architecture of the stems and vegetation (*Breceda, Ortiz & Scrosati, 2005*). Moreover, browsing may have even stronger effects on plant vital rates than defoliation for NTFPs harvesting since harvesting is generally restricted to a particular leaf or individual plant characteristics (*e.g.*, leaf size and shape) and, therefore, more selective than browsing by livestock that usually consume all available leaves.

Several studies have evaluated the effect of leaf harvesting on individual plant responses, especially in palms (*Svenning & Macia, 2002*; *Arango, Duque & Muñoz, 2010*; *Hernández-Barrios et al., 2012*; *Martínez-Ballesté & Martorell, 2015*). Few studies, however, have analyzed the interactive effects of two factors or more on the individual and demographic patterns of plants (*Berry et al., 2008*; *Mandle & Ticktin, 2012*; *Mandle, Ticktin & Zuidema, 2015*; *Sinasson & Shackleton, 2023*). These studies demonstrated interactions among two or more drivers on vital rates of individuals and on population dynamics. *Berry et al. (2008)* found significant interactions among substrate site, topographic position, human

management, and herbivory on the population dynamics of *Chamaedorea radicalis* in Mexico. Their results showed that herbivory reduced survival and fecundity in populations located on the forest floor, which in the absence of seed migration resulted in a projected decline of forest floor palms (sinks). With seed dispersal, however, the palms persisted, and the total population growth for both substrates was projected to be positive, indicating that seed dispersal from non-browsed palms on rock outcrops (sources) was sufficient to sustain, *C. radicalis* on the forest floor. Similarly, *Mandle, Ticktin & Zuidema (2015)* found that the population dynamics of *Phoenix loureiroi* palms in India are driven by interactive effects among fire, grazing, leaf harvest and abiotic conditions.

*Brahea aculeata* (Brandegee) H. E. Moore is an endemic palm of northwestern Mexico. The leaves are harvested for roof thatching and the production of handicrafts (baskets, *etc.*). Cattle also graze on the leaves of *B. aculeata* (*Joyal, 1996*; *Lopez-Toledo, Horn & Endress, 2011*). It is likely that defoliation, whether from harvesters or livestock, alone or combined, may affect the individual and demographic responses. This paper aims to understand the effects of grazing, leaf harvesting, and their potential interaction on individual traits and demographic patterns of *B. aculeata*. We expected that over three years of experiment (i) individuals of *Brahea aculeata* would tolerate browsing and low to moderate leaf harvest; however, if the leaf area is lost at high intensity, there would be negative impacts on individual traits. In terms of demographic impacts, we expected that (ii) harvest and livestock browsing would negatively affect survival, growth and reproduction and (iii) that the magnitude of these negative effects would increase over time. Furthermore, after three years, the cumulative leaf area loss would result in larger effects both on individual and demographic traits.

## MATERIALS & METHODS

### Study site

The study was conducted within the Área de Protección de Flora y Fauna Sierra de Alamos-Río Cuchujaqui (APFFSA-RC), a 92,890-ha federal protected area in the northern Mexican state of Sonora (27°12′30″–26°53′09″N and 109°03′00″–108°29′32″W). Within the APFFSA-RC, elevations range from 300 to 1,600 m and create a vegetation gradient ranging from tropical deciduous to pine-oak forest (*Haro, 2009*). Precipitation is highly variable with a mean of 650 mm, ranging from 190 and 1,120 mm/year (*Lopez-Toledo, Horn & Endress, 2011*). During this research, precipitation ranged from 360 (2011) to 472 (2014) mm, which is lower than the annual average recorded. In this region, the dry season is very pronounced and lasts up to 8 months (November to June). The mean annual temperature is 21.5 °C and ranges from 10 °C and 41 °C as minimum and maximum temperatures (*Haro, 2009*; *Lopez-Toledo, Horn & Endress, 2011*).

### Study species

*Brahea aculeata* (Brandegee) H. E. Moore (*Erythea aculeata* Brandegee) is a solitary-stemmed palm belonging to the family Arecaceae, subfamilia Coryphoideae (Fig. 1). It is endemic to northwestern Mexico (in the states of Sonora, Sinaloa, Chihuahua and Durango) and is found in patchy distributions in rocky soils from sunny mountains

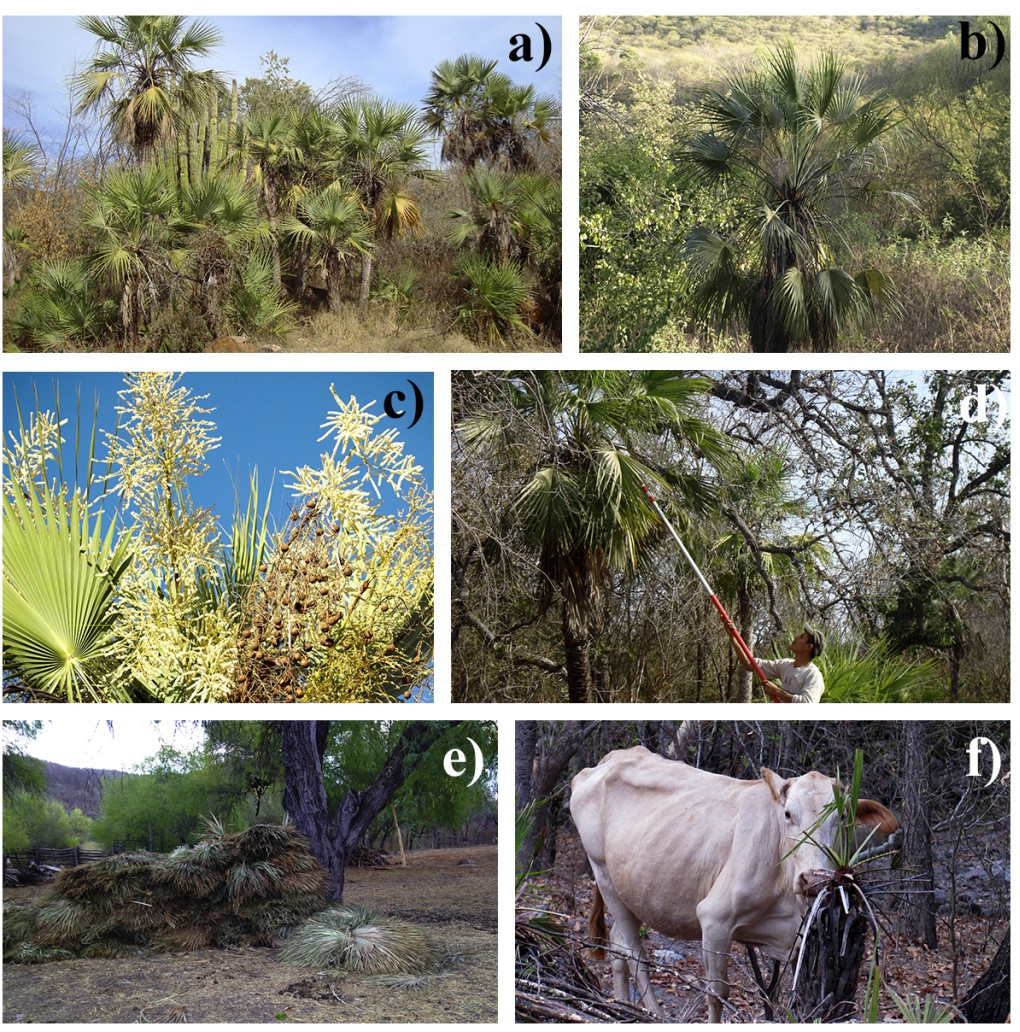

**Figure 1** *Brahea aculeata* palms in the tropical dry forest of Alamos, Sonora: (A) Patch of individuals, (B) isolated palm, (C) reproductive individual (D) leaf harvesting by local people, (E) harvested leaves, (F) livestock browsing on palms.

slopes to shadier areas along arroyos, rivers and canyon bottoms in tropical dry forest and lower oak and pine-oak woodlands (*Lopez-Toledo, Horn & Endress, 2011*). The species reaches ~10 m in height, is hermaphroditic, and individuals usually become reproductive after 1.0 m of height (*Quero, 2000*). The species flowers between March and May, and fruits ripen during May and November. It has a wide altitudinal distribution ranging from 320–1,500 m. *B. aculeata* appears on the IUCN Red List as ''vulnerable,'' as well as ''endangered'' in the Mexican Red List because of habitat loss and overexploitation of their leaves (*Quero, 1998*; *Felger, Johnson & Wilson, 2001*; *SEMARNAT, 2002*).

## Management of *Brahea aculeata*

Within the APFFSA-RC, *B. aculeata* is a very important species in terms of structure of the vegetation and as an NTFP (*Joyal, 1996*). Commercial leaf harvesting in the region has

been carried out for at least 50 years, with local residents developing traditional strategies for leaf harvest and palm management (*Joyal, 1996*; *Lopez-Toledo, Horn & Endress, 2011*). Within the APFFSA-RC, there are two leaf harvesting schemes: (i) that conducted by native people from inside the reserve, hereafter defined as "low harvest" and that (ii) conducted by non-indigenous people from larger cities or from coastal towns who use the leaves for thatching roofs of beach resorts, hereafter named "high harvest." The "low harvest" scheme usually involves harvesting only mature leaves, and always only the two youngest leaves, and spear leaves are left. In contrast, the "high harvest" involves a more aggressive harvesting scheme, cutting all available leaves and spears. Depending on demand for the product, some areas may be harvested every 6 months or every year. Furthermore, within the reserve, cattle ranching is very common, with the cattle placed in different plots of land of 30–50 ha within a farm and rotated every 2–3 months (*Quisehuatl-Medina et al., 2020*). During the dry season when many of the deciduous trees have dropped their leaves, *B. aculeata* leaves may be consumed by cattle, which may damage palm individuals <250 cm high (*Lopez-Toledo, Horn & Endress, 2011*). Leaf harvesting is carried out on juveniles and adult plants because they produce good-sized leaves for both harvesting schemes, while cattle activity may affect all life stages through grazing, trampling and soil compaction (*Quisehuatl-Medina et al., 2020*). Leaf harvesting occurs in January, which coincides with the template climate and based on traditional knowledge results in the highest leaf quality for weaving and thatching.

## Sampling design

To evaluate the effect of cattle grazing and leaf harvesting on *Brahea aculeata,* in 2011 we established six permanent plots, each measuring 75 × 25 m (0.1875 ha), for a total sampling area of 1.125 ha. All *B. aculeata* individuals within the plots were used in the experiment. Two treatments were randomly assigned to (1) grazing and (2) non-grazing plots. For the former treatment, unrestricted livestock grazing was allowed, while the latter plots were fenced to prevent cattle access. To simulate the local management, cattle were allowed within the plot for about 3 months (April to June) each year.

Each plot was then divided into three subplots of 25 × 25 m (625 m$^2$) where three regional harvest schemes were randomly assigned and applied annually in January each year: (1) no harvest, (2) low harvest and (3) high harvest. Using gardening pruners, we applied these harvesting schemes for each palm ≥ 10 cm in height and quantified the total number of leaves and harvested leaves. Harvesting treatments resulted in a gradient from 0 to 100% of harvested leaves in both grazing treatments. We obtained the necessary permits from Dirección General de Vida Silvestre-Secretaria de Medio Ambiente y Recursos Naturales to collect leaves (DGVS/01991/10 and DGVS/00837/14).

Annual censuses were conducted from January 2011 to January 2014. Each palm was measured in January at each census, and we marked the second youngest leaf, which represented a fully developed leaf, with paint; thus, we were able to measure new leaf production each year. Using a measuring tape, we measured or counted the following variables for each palm: (i) length of stem (from the base of the trunk to the base of the youngest leaf), (ii) leaf production (number of leaves produced above the marked leave),

(iii) lamina length (of the marked leaf and hereafter defined as "leaf length"), (iv) petiole length (of the marked leaf), (v) percentage of leaves harvested, (vi) survival (alive or dead) and (vii) number of fruits (except for 2011 to 2012). We classified individuals into one of three size classes: (a) *Juveniles*, individuals of 10.1–100 cm and base diameter > five cm; (b) *Adult I*, which comprised individuals of 100.1–250 cm) and (d) *Adult II*, individuals > 250 cm.

Overall, we marked and monitored a total of 1,194 individuals during the three-year study period, with 197–228 individuals per plot. This included the following number of individuals per treatment: 120–125 juveniles, 45–63 Adult I and 32–40 Adult II.

## Data analysis

We included seven different response variables including three at individual and four at demographic level, respectively. The individual traits analyzed were related to leaf area and included leaf production (LP), leaf length (LL) and petiole length (PL). The demographic rates analyzed included mortality (M), stem growth (SG), probability of reproduction (PR) and fruit production (FP). To assess the responses of browsing and leaf harvesting on *Brahea aculeata* at the individual and demographic level, we developed mixed models considering the following explanatory variables: Time ('T' with three levels: 2012, 2013 and 2014), Grazing (included as a categorical variable 'Gr' with the levels Grazing and Non-Grazing), Harvest ('H' expressed as a proportion of harvested leaves/total leaves varying from 0 to 100% harvest). We also included the interaction term Harvest: Time to consider the cumulative effects of harvesting through time, which we expected would become stronger through the monitoring. For all analyses, these were used as fixed factors. To reflect the nested design of the experiment, the "plot/subplot" of the browsing/harvesting treatments and the repeated measurements of individuals through time, we used mixed models, which can include these terms as random effects (*Pinheiro & Bates, 2000*; *Bates et al., 2015*). All the analyses were conducted using linear mixed-effects model (LMM) for continuous response variables (*e.g.*, leaf length, petiole length and stem growth) and the generalized linear mixed model (GLMM) for variables that were counts (*e.g.*, leaf production and fruit production) and binomials (*e.g.*, probability of reproduction). We did not conduct any statistical analysis for mortality given the very low number of dead individuals (nine in total) found over the study period. For LMM analyses, when required, response variables were log(x) or log (x + 1) transformed to meet normality criteria (*Crawley, 2012*). For GLMM analyses, we used Poisson error for counts and binomial error for binomial variables. The model presented for each response variable represents the full model including all terms mentioned above. The trend line presented in the results was plotted based on the coefficients of these models; see Supplementary Files. All analyses were carried out using the *lme4* package 1.1-21 version (*Bates et al., 2015*) in the R program version 3.5.2 (*The R Foundation for Statistical Computing, 2018*). To test significance of explanatory variables, we used the *lmerTest* package version 3.1-0 for LMM (*Kuznetsova, Brockhoff & Bojesen-Christensen, 2016*) and the parametric bootstrap method available in the *pbkrtest* package ver. 0.4-7 for GLMMs (*Halekoh & Højsgaard, 2014*).

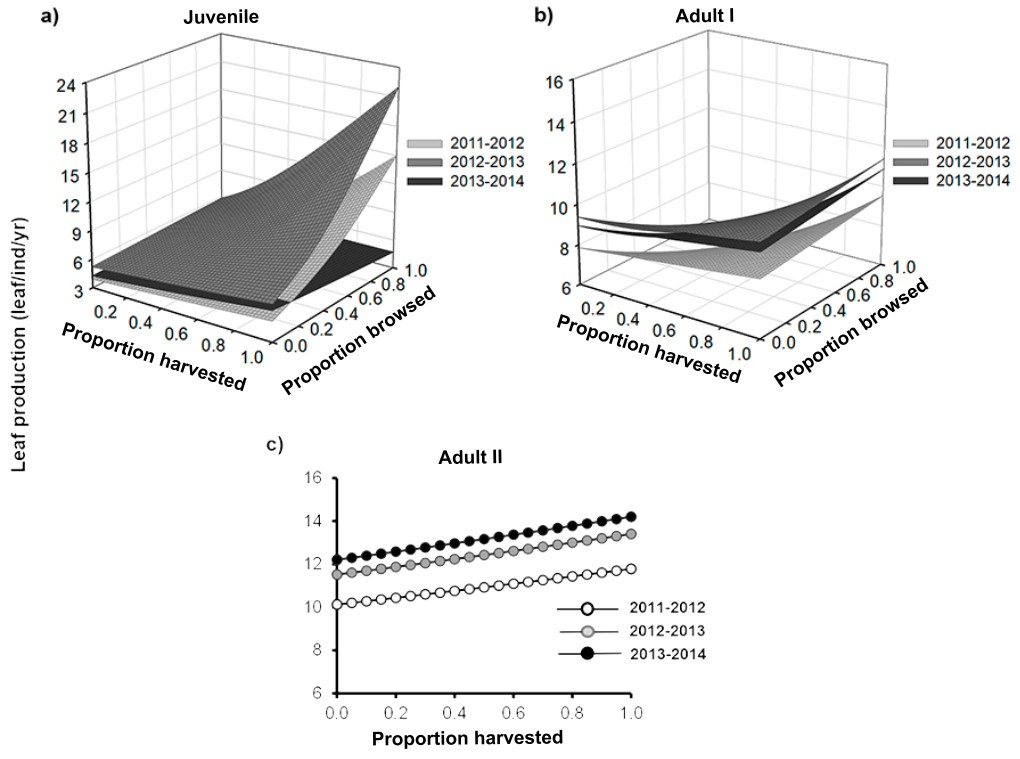

**Figure 2  Leaf production of *Brahea aculeata* individuals subjected to different intensities of browsing and harvesting of leaves in the tropical dry forest of Alamos, Sonora: (A) Juveniles, (B) Adult I and (C) Adult II.** In the figure the different surfaces/trend lines represent the values predicted by the statistical model for each year of sampling. Note the differences in the $Y$-axis scale.

## RESULTS

### Individual traits

Overall, we found that the effects of leaf harvesting were stronger than those of browsing ($\chi^2 = 45.3$, $p < 0.001$). Leaf harvest had significant effects on the three individual traits analyzed. Palm performance, however, was similar between browsing treatments (Fig. 2). Furthermore, the null effects of browsing were similar for the three size classes analyzed (Tables 1–3).

Leaf production ranged from 0 to 35 leaves/yr and differed among size categories (Fig. 2). Overall, the lowest leaf production rate was recorded for juveniles (mean $\pm$ SE: 5.3 $\pm$ 0.05 leaves/yr), while the highest was for the adult II size-class (mean $\pm$ SE: 12.8 $\pm$ 0.3 leaves/yr) (Fig. 2). For all the three size classes, we found that palms showed a positive response to harvesting, that is, the rate of leaf production significantly increased with an increase in the proportion of leaves harvested (Fig. 2; Table 1). Although this relationship was positive for the three years, the slope varied over time, increasing in the second year, and the declining in the third year (Figs. 2A and 2B; Table S1).

Leaf length also differed among palm size class, and juveniles had the smallest leaves (mean $\pm$ SE: 48.3 $\pm$ 0.2 cm), while Adult I and Adult II size classes had 1.5–1.2 times

**Table 1   Results of generalized linear mixed-effects models (GLMMs) used to assess the cumulative effects of browsing and harvesting of leaves on leaf production of *Brahea aculeata* in the tropical dry forest of Alamos, Sonora.** The terms tested in the models were the time (T), browsing proportion (Br), harvesting proportion (H) and the interactions among these terms (Br:H). The statistics are provided $F$ and $\chi^2$, degrees of freedom in brackets ($gl$) and $P$ value with based in maximum likelihood ratio test; ns indicates that there was no significant effect; (−) indicates that the factor was not significant and therefore it was removed from the model (See Fig. 1).

| | Leaf production | | | | | | | |
|---|---|---|---|---|---|---|---|---|
| Factors | Seedlings | | Juveniles | | Adults I | | Adults II | |
| | $F/LRT\,\chi^2_{(gl)}$ | $P$ | $F/LRT\,\chi^2_{(gl)}$ | $P$ | $F/LRT\,\chi^2_{(gl)}$ | $P$ | $F/LRT\,\chi^2_{(gl)}$ | $P$ |
| Time (T) | − | − | $41.8_{(2,1987)}$ | <0.001 | $40.5_{(2)}$ | <0.001 | $19.9_{(2)}$ | <0.001 |
| Browsing (Br) | − | − | $6.9_{(1,1987)}$ | 0.008 | $12.5_{(1)}$ | <0.001 | | |
| Harvesting (H) | NA | NA | $97.5_{(1,1987)}$ | <0.001 | $5.4_{(1)}$ | 0.01 | $4.5_{(1)}$ | 0.03 |
| T:Br | − | − | $0.7_{(2,1985)}$ | ns | − | − | | |
| T:H | | | $3.8_{(2,1987)}$ | 0.02 | − | − | − | − |
| Br:H | | | $6.8_{(1,1987)}$ | 0.008 | $3.9_{(1)}$ | 0.04 | | |
| T:Br:H | | | $5.4_{(2,1985)}$ | 0.004 | − | − | | |

**Table 2   Results of linear mixed-effects models (LMM) used to assess the cumulative effects of browsing and harvesting of leaves on leaf length of *Brahea aculeata* in the tropical dry forest of Alamos, Sonora.** The terms tested in the models were the time (T), browsing proportion (Br), harvesting proportion (H) and the interactions among these terms (Br:H). The statistics are provided $F$ and $\chi^2$, degrees of freedom in brackets ($gl$) and $P$ value with based in maximum likelihood ratio test; ns indicates that there was no significant effect; (−) indicates that such factor was not significant and therefore it was removed from the model (See Fig. 2).

| | Leaf length | | | | | | | |
|---|---|---|---|---|---|---|---|---|
| | Seedlings | | Juveniles | | Adult I | | Adult II | |
| Factors | $F/LRT\,\chi^2_{(gl)}$ | $P$ | $F/LRT\,\chi^2_{(gl)}$ | $P$ | $F/LRT\,\chi^2_{(gl)}$ | $P$ | $F/LRT\,\chi^2_{(gl)}$ | $P$ |
| Time (T) | − | − | $28.1_{(2,1982)}$ | <0.001 | $68.9_{(2)}$ | <0.001 | $22.8_{(2)}$ | <0.001 |
| Browsing (Br) | − | − | $1.2_{(1,1982)}$ | ns | $11.2_{(1)}$ | <0.001 | | |
| Harvesting H) | − | − | $21.3_{(1,1982)}$ | <0.001 | $0.7_{(1)}$ | ns | − | − |
| T:Br | − | − | $3.5_{(2,1982)}$ | <0.02 | − | − | | |
| T:H | − | − | $17.8_{(2,1982)}$ | <0.001 | − | − | − | − |
| Br:H | − | − | $6.3_{(1,1982)}$ | 0.01 | $6.3_{(1)}$ | 0.01 | | |
| T:Br:H | | | − | − | − | − | | |

larger leaves, respectively (Fig. 3; Table S1). We detected effects of browsing only for Adult I individuals but not for juveniles (Table 2). Harvesting was significant for juveniles and Adult I individuals. In the first year, harvesting produced positive effects with larger leaves at higher harvesting levels (slope = 5.7 cm/proportion of leaves harvested). For the second year, leaf size decreased with harvesting, and, for the third year, this relationship became negative (slope = 0.5 cm/proportion of leaves harvested, Table S2). For the Adult I and Adult II size classes, harvesting did not produce changes in leaf length during the first and second year, but for the third the relationship between harvest and leaf length was negative (Fig. 3; Table 2 and S2).

Petiole length was a very sensitive trait, and generally high harvesting intensities had a positive effect on palm petiole length during the two first years, with an increase of up to 55% in length as compared to controls. For the third year, however, harvest decreased

**Table 3  Results of linear mixed-effects models (LMM) used to assess the cumulative effects of browsing and harvesting of leaves on petiole length of *Brahea aculeata* in the tropical dry forest of Alamos, Sonora.**  The terms tested in the models were the time (T), browsing proportion (Br), harvesting proportion (H) and the interactions among these terms (Br:H). The statistics are provided $F$ and $\chi^2$, degrees of freedom in brackets (*gl*) and $P$ value with based in maximum likelihood ratio test; ns indicates that there was no significant effect; (−) indicates that such factor was not significant and therefore it was removed from the model (See Fig. 3).

| | Petiole length | | | | | | | |
| --- | --- | --- | --- | --- | --- | --- | --- | --- |
| | *Seedlings* | | *Juveniles* | | *Adult I* | | *Adult II* | |
| Factors | $F/LRT\,\chi^2_{(gl)}$ | $P$ | $F/LRT\,\chi^2_{(gl)}$ | $P$ | $F/LRT\,\chi^2_{(gl)}$ | $P$ | $F/LRT\,\chi^2_{(gl)}$ | $P$ |
| Time (T) | $15.2_{(2)}$ | <0.001 | $139.5_{(2,1988)}$ | <0.001 | $175.09_{(2,891)}$ | <0.001 | $9.04_{(2,283)}$ | <0.001 |
| Browsing (Br) | − | − | $0.7_{(1,1988)}$ | ns | $6.2_{(1,891)}$ | 0.01 | | |
| Harvesting (H) | | | $5.1_{(1,1988)}$ | 0.02 | $2.3_{(1,891)}$ | ns | $13.6_{(1,283)}$ | <0.001 |
| T:Br | − | − | $9.6_{(2,1988)}$ | <0.001 | − | − | | |
| T:H | − | − | − | − | $4.5_{(2,891)}$ | 0.01 | $6.7_{(2,283)}$ | 0.001 |
| R:H | | | − | − | − | − | − | − |
| T:Br:H | | | − | − | − | − | − | − |

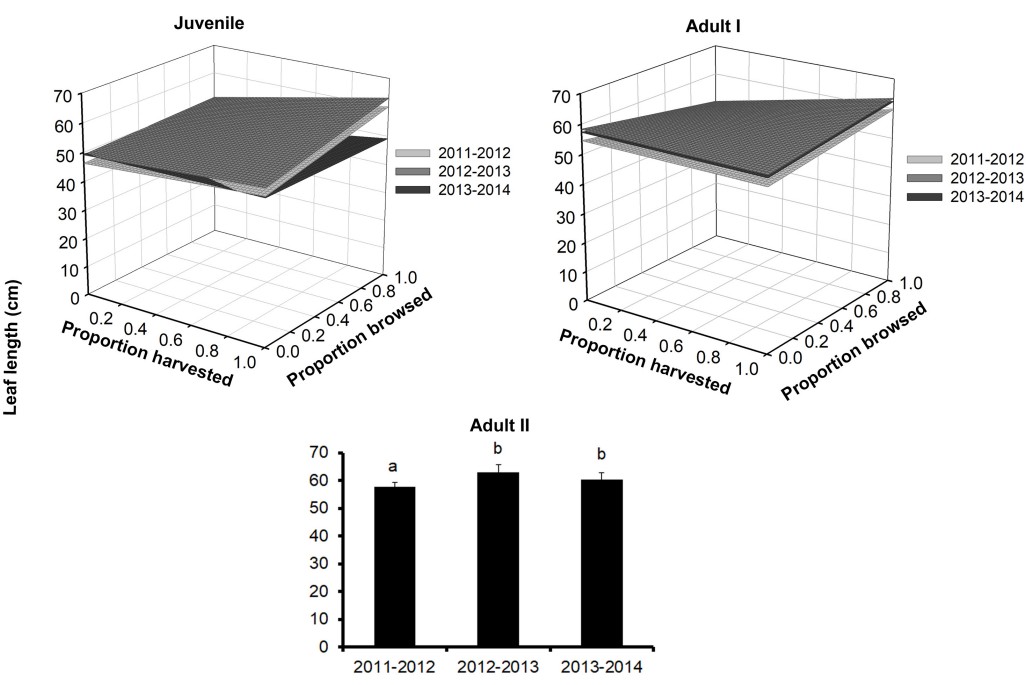

**Figure 3  Leaf length of *Brahea aculeata* individuals subjected to different intensities of browsing and harvesting of leaves in the tropical dry forest of Alamos, Sonora: (Top left) Juveniles, (Top right) Adult I and (Bottom) Adult II.**  In the Juvenile and Adult I figures, the different surfaces represent the values predicted by the statistical model for each year of sampling. In Adult II, the letters represent significant differences for sampling years.

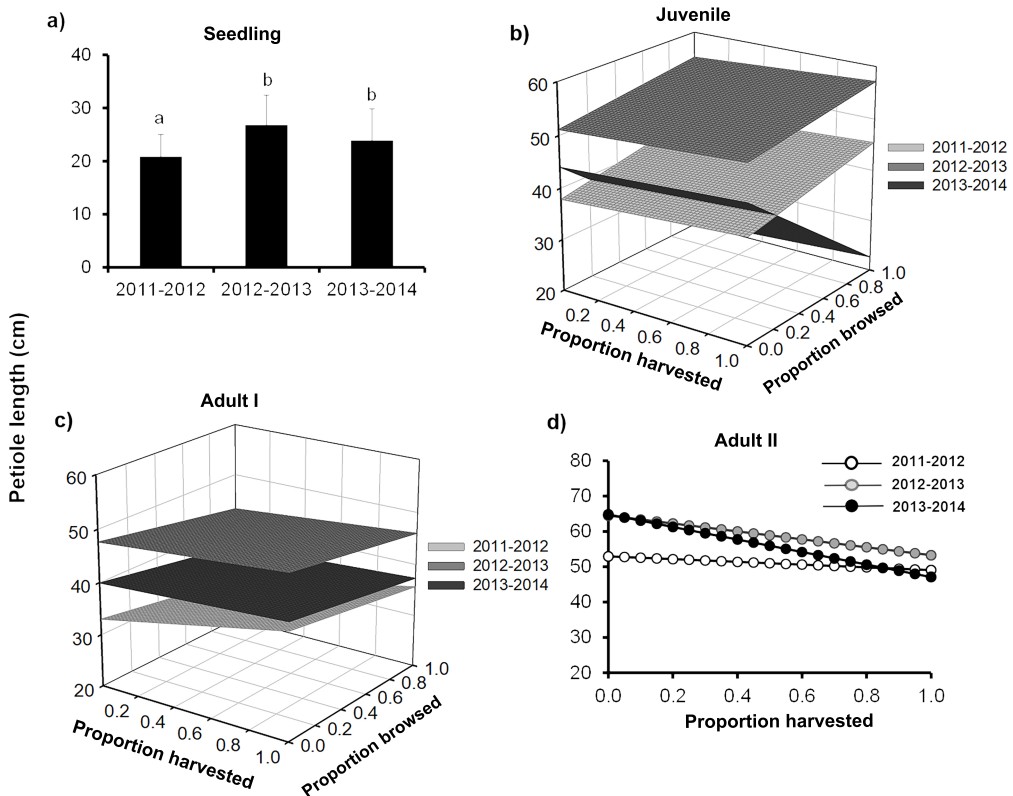

**Figure 4  Petiole length of *Brahea aculeata* individuals subjected to different intensities of browsing and harvesting of leaves in the tropical dry forest of Alamos, Sonora: (A) Seedlings, (B) Juveniles, (C) Adult I and (D) Adult II.**  In (A) the letters represent significant differences for years of sampling; while in (B), (C) and (D) the different surfaces/trend lines represent the values predicted by the statistical model for each year of sampling. Note the differences in the $Y$-axis scale.

petiole length (Table 3; Fig. 4). We did not find any effect of browsing on petiole length (Table 3).

## Demographic rates

Similar to the individual traits, we did not find a significant effect of browsing on any of the demographic rates of *Brahea aculeata*. Since very few individuals died over the study period (nine individuals–seven juveniles and two Adult I), we did not conduct any statistical tests. We found a large range of variation in stem growth, ranging from 0 and 25 cm yr$^{-1}$. This was mainly due to palm size, where individuals in the Adult I and Adult II size classes showed the highest and lowest growth respectively (Fig. 5). There was no significant difference in stem growth among plants in the browsed and non-browsed plots. Stem growth was higher in the third year than in the first, and increased as a function of the proportion of leaves browsed; but decreased with the proportion of leaves harvested in the second year. The effects of harvesting on stem growth were negative during the three years (Table 4; Figs. 5A, 5B).
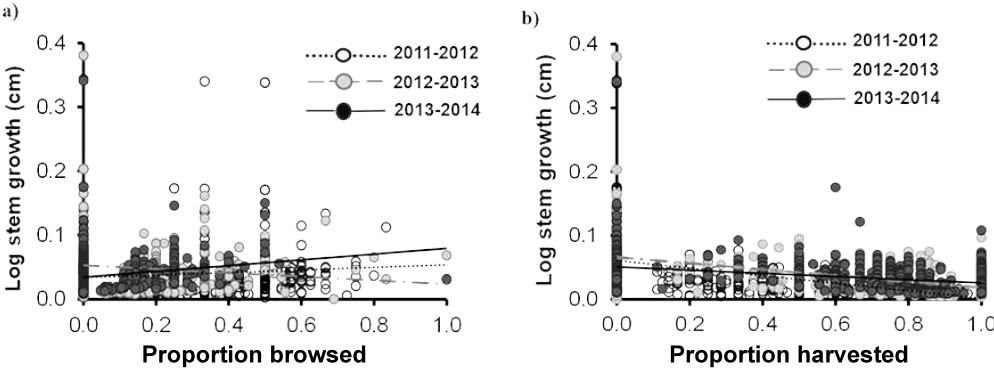

**Figure 5** Stem growth for *Brahea aculeata* individuals subjected to different intensities of browsing and harvesting of leaves in the tropical dry forest of Alamos, Sonora: (A) stem growth and (B) stem growth. In both figures the circles represent the observed values and the different colors represent the sampling years. The trend lines represent the values predicted by the model for each year of sampling. Note the logarithmic scale of the vertical axis in the plot.

**Table 4** Results of linear model using generalized least squares (GLS) used to assess the cumulative effects of browsing and harvesting of leaves on stem growth *Brahea aculeata* in the tropical dry forest of Alamos, Sonora. The terms tested in the models were time (T), browsing proportion (Br), harvesting proportion (H), stem length (SL) and the interactions among these factors. The statistics are provided ($\chi^2$, degrees of freedom in brackets and *P* value based in maximum likelihood ratio test; ns indicates no significant effect of the term and –indicates that such factor was not significant and therefore it was removed from the model. Three and four way interactions (SL:T:Br , SL:T:H, SL:Br:H, T:Br:H, SL:T:Br:H) were not significant and were removed from the model and therefore they are not included in the table (See Fig. 4).

| | Stem growth | |
|---|---|---|
| **Factors** | **$\chi^2$** | **P** |
| Stem length (SL) | $253.5_{(1)}$ | <0.001 |
| Time (T) | $23.5_{(2)}$ | <0.001 |
| Browsing (Br) | $8.9_{(1)}$ | <0.01 |
| Harvesting (H) | $1.1_{(1)}$ | ns |
| SL:T | $7.27_{(2)}$ | 0.02 |
| SL:Br | – | – |
| SL:H | – | – |
| T:Br | $7.08_{(2)}$ | 0.02 |
| T:H | $7.08_{(2)}$ | 0.02 |
| Br:H | – | – |

Palms subjected to leaf area loss showed apparently contrasting effects on attributes related to reproduction. In general, the probability of reproduction increased with leaf area loss, while the number of fruits produced decreased with harvesting intensity. The positive effect of harvesting on the probability of reproduction was especially evident for Adult I individuals, which also showed an increase over time in the probability of reproduction. Harvesting intensities >20% increased the probability of reproduction, and, after 40% harvest, the probability reached its maximum (Fig. 6A, Table 5). Moreover, the

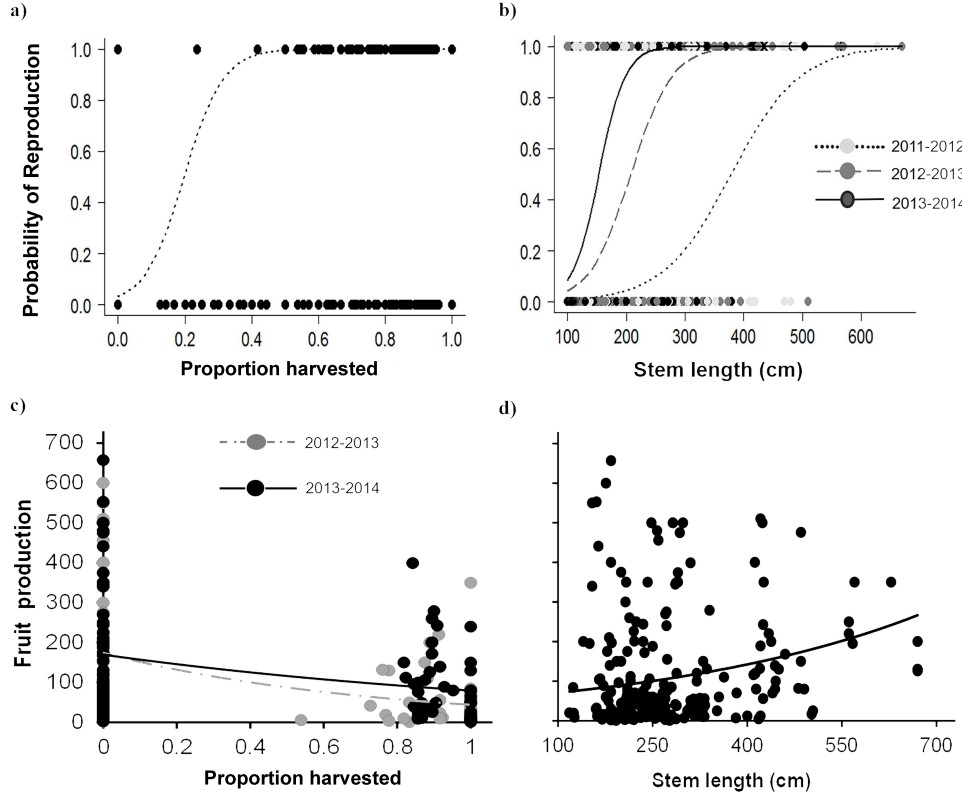

**Figure 6** **Reproductive attributes for *Brahea aculeata* under different harvesting intensities in the tropical dry forest of Alamos, Sonora.** (A) Probability of reproduction as a function of harvesting, (B) probability of reproduction as a function of size and (C) fruit production as a function of harvesting, (D) fruit production as a function of size. In all the figures the circles represent the observed values, the lines represent the values predicted by model and, the different colors in circles and lines represent the sampling years, in both figures lines represent the values predicted by the statistical model.

probability of reproduction varied positively with the size of individuals, and with larger individuals having higher reproduction probabilities. However, this also in the third year, as the probability of reproduction increased in smaller palms (Table 5; Fig. 6B).

Fruit production varied largely among individuals from two to 657 fruit/individual, and a noteworthy variation among years was also recorded (Fig. 6C) This variation was partly explained by the proportion of leaves harvested, time and stem length, but browsing had no effect (Table 5). Harvesting had a negative effect on fruit production, and, for the most part, high intensity harvesting reduced fruit production by about 50% of (Fig. 6C). Although we found high variation in fruit production, fruit production increased as a function of palm size (Fig. 6D).

## DISCUSSION

Overall, we found that *Brahea aculeata* is largely resilient to browsing and leaf-harvesting. Based on our results, *B. aculeata* can be considered a species that tolerates high levels of defoliation and/or browsing for 2 years, but not likely longer.

**Table 5  Results of generalized linear mixed-effects models (GLMMs) or generalized linear mixed models using AD model Builder (glmmADMB) used to assess the cumulative effects of browsing and harvesting of leaves on reproductive attributes *Brahea aculeata* in the tropical dry forest of Alamos, Sonora.** The terms tested in the models were time (T), browsing proportion (Br), harvesting proportion (H), stem length (SL) and the interactions among these factors. We used glmer for the reproduction probability and glmmadmb for the fruit number. The statistics are provided ($\chi^2$, degrees of freedom in brackets and *P* value based in maximum likelihood ratio test; ns indicates no significant effect of the term and –indicates that such factor was not significant and therefore it was removed from the model). Three and four way interactions (SL:T:Br , SL:T:H, SL:Br:H, T:Br:H, SL:T:Br:H) were not significant and were removed from the model and therefore they are not included in the table (See Fig. 5).

| Factors | Probability of reproduction | | Fruit production | |
|---|---|---|---|---|
| | $\chi^2$ | *P* | $\chi^2$ | *P* |
| Stem length (SL) | $20.8_{(1)}$ | <0.001 | $17.68_{(1)}$ | <0.001 |
| Time (T) | $1.01_{(2)}$ | ns | $6.9_{(2)}$ | 0.03 |
| Browsing (Br) | – | – | – | – |
| Harvesting (H) | $8.59_{(1)}$ | 0.003 | $10.18_{(2)}$ | 0.006 |
| SL:T | $26.06_{(2)}$ | <0.001 | – | – |
| SL:Br | – | – | – | – |
| SL:H | – | – | – | – |
| T:Br | – | – | – | – |
| T:H | – | – | $3.9_{(1)}$ | 0.04 |
| Br:H | – | – | – | – |

## Individual traits

To our knowledge, few studies have tested the effects of different drivers and their interactions on non-timber forest products (NTFPs) (*e.g.*, *Endress, Gorchev & Noble, 2004*; *Mandle & Ticktin, 2012*; *Sinasson & Shackleton, 2023*), yet these experimental designs are necessary to simulate real management practices where NTFPs are exposed to other disturbance factors in addition to harvest. Our study contributes to this goal by documenting the effect of multiple drivers on both individuals and populations. During the first two years, we found that loss of leaf area had limited effects on *Brahea aculeata,* and individuals were able to recover from defoliation. After 3 years of defoliation, however, individuals subjected to both high browsing and high harvesting rates were more negatively affected. This is consistent with research on *Phoenix loureiroi*, the mountain date palm, where grazing and harvest reduced growth (*Mandle & Ticktin, 2012*).

Our experimental management of *Brahea aculeata* only negatively affected one of the three individual traits analyzed (petiole length). Although petiole size is not relevant for the management of the species, it is an architectural attribute important for structural support (*Niinemets et al., 2004*). The changes we found may help to understand the low or null effect on leaf size and maybe indicate possible trade-offs among functions such as light acquisition and support structures (*Niinemets et al., 2004*). Leaf size and leaf production generally responded positively, to both harvest and browsing and we did not find any effect on the total number of leaves per plant. Although leaf production and leaf size showed a reduction in the three size-categories, this was not lower than the first-year values. Therefore, this species may be considered resilient: capable of recovering from damage

caused by defoliation even at high leaf area loss intensities (*Walker, Kinzig & Langridge, 1999*; *Lopez-Toledo et al., 2012*), at least in the short term. This coincides with findings for other palms species subject to low defoliation intensity treatments of infrequent harvesting (*Martínez-Ballesté, Martorell & Caballero, 2008*; *Hernández-Barrios et al., 2012*; *Mandle, Ticktin & Zuidema, 2015*; *Pulido & Coronel-Ortega, 2015*). *Mandle, Ticktin & Zuidema (2015)* studying *Phoenix loureiroi* in India, reported that this species is resilient to low (~15%) harvest rates with an annual harvest scheme.

In the case of *Brahea aculeata*, the increase in leaf production and leaf size with harvest and browsing is likely due to an increased allocation of resources to these attributes. This response can be considered an overcompensatory response (*Bazzaz, Ackerly & Reekie, 2000*; *Anten & Ackerly, 2001*; *Anten, Martinez-Ramos & Ackerly, 2003*), which has been explained as follows: (i) a result of the mobilization of resources from other plant structures, such as stems or roots that store nonstructural carbohydrates and nutrients; (ii) a reallocation at the expense of some functions such as reproduction or (iii) a response caused by an adjustment in the photosynthesis rate in the remaining leaves (*Endress, Gorchov & Berry, 2006*; *Lopez-Toledo et al., 2012*; *Martínez-Ballesté & Martorell, 2015*). For example, defoliated *Chamaedorea elegans* palms allocated more resources to lamina growth at the expense of the other plant structures, especially those related to reproduction (*Anten, Martinez-Ramos & Ackerly, 2003*). Studies of other palm species under more intensive management schemes (semiannual harvest and high harvesting intensities) have also found a higher leaf production rate (*Martínez-Ballesté, Martorell & Caballero, 2008*; *Coronel & Pulido, 2010*). In a study of two species of *Sabal* (*S. yapa* and *S. mexicana*) from the Mexican Yucatan Peninsula, *Martínez-Ballesté & Martorell (2015)* also concluded that adult individuals could compensate for leaf production even at higher harvest intensities (twice each year) for 2 years. As part of the experiment, however, one to three young leaves per palm were left, which allowed them to recover. In our case, intensive management included the harvest of all leaves and spears, and, even under this scenario, we did not find negative effects on *Brahea aculeata* for the first two years, which again highlights the resilience of this species.

With the exception of a few cases, most of the defoliation studies of palms have been short-term such 1 or 2 years (*Martínez-Ballesté, Martorell & Caballero, 2008*; *Duarte & Montúfar, 2012*; *Pulido & Coronel-Ortega, 2015*). To evaluate the effect of leaf area loss, especially in the context of the sustainability of non-timber forest products, it is important to conduct long-term studies, as short-term studies may miss interannual variability and may not detect the cumulative effects of leaf area loss. In the case of *Brahea aculeata*, experimentally conducting real management practices over 3 years showed an initial increase in all the response variables analyzed, with the highest levels in the second year, but then a drop in the third year, especially for juveniles. This decline may indicate the cumulative effects of three years of browsing and harvesting. In other palm species, multiple defoliation events and intensive harvesting lead to a depletion of reserves (*Martínez-Ramos, Anten & Ackerly, 2009*). For example, a defoliation experiment on *Chamaedorea elegans* led to a large reduction in leaf and reproductive traits, especially at high harvesting intensities (*Lopez-Toledo et al., 2012*).

For *Brahea aculeata*, in the third year of the experiment, the effect of harvest and browsing on leaf attributes was greater on intermediate sizes than on smaller or larger individuals. This was due to the interactive effects of browsing and harvesting. The intermediate-sized palms (10–250 cm height) can be easily harvested, and the leaves are more accessible to livestock. In contrast, smaller and larger individuals (seedlings and adult II) were only subjected to either browsing or harvesting but not both. The small effect of browsing and harvesting on adult II leaf attributes is remarkable given that adults need to allocate resources for reproductive structures. This indicates the high resilience of these size classes. In large adults, the stored resources are, in general, more abundant, which likely explains the lower impacts we found (*Anten, Martinez-Ramos & Ackerly, 2003*; *Boege, 2005*). Several studies, however, have found that plants have thresholds at which they can tolerate some degree of disturbance, but once this is exceeded, accumulated reserves are reduced or depleted, and individuals are no longer able to compensate for the damage (*Anten & Ackerly, 2001*; *Staffan & Méndez, 2005*). It is likely that in the case of *Brahea aculeata* our results indicate that we were just about to reach this threshold. Thus, we recommend future studies would benefit from experiments that aim to identify this threshold, by either increasing the frequency of harvesting or the number of harvests. This may allow scientist to offer recommendations more specific for a management plan. However, for *Brahea aculeata*, it will remain open for future studies.

## Demographic patterns

Other palms such as *Phoenix loureiroi,* which is an NTFPs from Indian tropical dry forests, have been found to be resilient to low levels of leaf area loss resulting from the combination of multiple interacting factors (fire, harvesting and grazing). Nevertheless, chronic and higher levels of these multiple factors have severe negative consequences for populations (*Mandle & Ticktin, 2012*). *Martínez-Ballesté & Martorell (2015)* studying the *Sabal yapa* palm from tropical dry forest ecosystems, also found that the species is resilient to harvest. In this case, *S. yapa* was resilient to any leaf harvesting regime and could maintain positive demographic rates. For *Brahea aculeata*, we concluded that this species is also resilient but only under a management that includes 2 years of annual harvesting, even high levels of harvesting. If this management continues for a third year, however, resource allocation patterns appear to change considerably, and some vital rates decline. There are other species such as *Chamaedorea (C. elegans, C. ernesti - augustii* and *C. oblongata)*, which have been reported to be susceptible to leaf harvesting (*Hernández-Barrios et al., 2012*; *Lopez-Toledo et al., 2012*). Although these species grow in different ecosystems (understory palms from humid tropical forests) than *B. aculeata*, their resource allocation patterns and effects of harvesting may be used to compare with our results. For some *Chamaedorea* species, several authors have found that high harvesting intensities (>66%) strongly affected demographic rates and population dynamics, even after few events (*Martínez-Ramos, Anten & Ackerly, 2009*; *Hernández-Barrios et al., 2012*; *Lopez-Toledo et al., 2012*). The difference may be due in part to environmental conditions and the frequency of defoliation. In these studies, *Chamaedorea* harvesting was performed biannually, whereas in our study, harvesting was conducted only once a year (to simulate one of the most frequent practices). Furthermore,

browsing was conducted during a short season of the year (May to July). Therefore, during the first two years of monitoring, it is likely that the time (one entire year) between events of defoliation may be sufficient for *B. aculeata* to recover from the leaf area lost. In addition, there may be differential responses between an understory palm such as *Chamaedorea* to a canopy species such as *Brahea*.

In previous studies, the first defoliation event can cause an increase in the probability of reproduction, flowers or fruits (*Martínez-Ramos, Anten & Ackerly, 2009*). The cumulative effects of multiple defoliation events, however, can lead to declines in reproductive output (*Endress, Gorchev & Noble, 2004*; *Zuidema, De Kroon & Werger, 2007*; *Hernández-Barrios et al., 2012*). This reduction has been explained based on the amount of carbohydrates and the carbon gain by photosynthesis, which is decreased by defoliation (*Klinkhamer et al., 1992*; *Boege, 2005*). Although leaf harvesting for *Brahea aculeata* increased the probability of reproduction, fruit production was two to three times lower as compared to control palms. Thus, it is likely that the frequency and intensity of defoliation applied to *Brahea aculeata* was sufficient to reach that threshold, indicating that the stored resources for reproduction became reduced. For further studies, it will be interesting to explore whether more frequent harvests or chronic harvesting over the long term affect the reproductive success or the fate of descendants. In the Alamos area, the species has been intensively harvested for at least 50 years. Given low recruitment of seedlings observed in the field, (may be due to cattle activity), it is likely that the regeneration may have been affected. This is an unanswered question and further studies related to reproductive success and germination are necessary.

*Brahea aculeata* is a long-lived species, and, therefore, low mortality is expected for adults. We found high survival rates and no impact on survival due to browsing or harvesting. For other harvested palm species, effects on mortality have been found, but this may be due to the greater frequency and intensity of harvesting or differences in the natural history of the species (*Endress, Gorchov & Berry, 2006*; *Martínez-Ramos, Anten & Ackerly, 2009*; *Mandle & Ticktin, 2012*).

Finally, the positive effect of browsing on growth may be an indirect effect of livestock activity. The cover of herbaceous plants was extremely high in the non-browsing plots, while in the browsing plots the livestock either trampled or ate the vegetation. This may eliminate the competition for resources, such as light, water or nutrients and make them available for palms (*Herrera, 1995*; *Heckel et al., 2010*).

## Implications for management

Palms are ecologically and economically important in many tropical regions, and especially the leaves are intensively used for different purposes, including thatching roofs and handicrafts, which generate important income for local people (*Arango, Duque & Muñoz, 2010*; *Coronel & Pulido, 2010*; *Duarte & Montúfar, 2012*). Our results indicate that harvest of *Brahea aculeata* may be sustainable under a two-year management cycle that includes annual harvesting at moderate and even high leaf harvesting intensity. However, any intent of management for 3 or more years should be completely avoided, and the rest period needed before harvest can resume is currently unknown. For *B. aculeata* and in

general for another NTFPs, it is will be important to continue to explore and identify responses and resilience to different frequencies of harvesting, intensity (proportion of product harvested), number of events of harvesting and incorporate periods of resting (non-harvesting). These kinds of experiments may contribute to valuable insights for long-term management.

Our results have important implications for the management of the species, as to date *B. aculeata* is a red listed species under Mexican and international norms, and harvest is illegal. In addition, within our study area, management on some properties may be even more intensive, with semiannual harvest (*Lopez-Toledo, Horn & Endress, 2011*). Many residents depend on leaf harvest of this species, and, therefore, we believe our results may contribute basic ecological sound information for the sustainable management of the species. These results may be implemented and contribute to an economically important activity. Note that characteristics such as leaf production rate and leaf length have direct implications for the use of many palms. Many handicrafts and roofs made of palm require only longer leaves (*Pulido & Caballero, 2006*; *Pulido & Coronel-Ortega, 2015*). Therefore, in this specific case, harvesting and browsing can be considered as positive—at least in the short term—leading to an increase in the length of leaves.

## CONCLUSIONS

Based on our experiment in field condition in northwestern Mexico, *Brahea aculeata* demonstrated considerable resilience to simultaneous defoliation and browsing events; however, it was able to compensate for these effects only during the initial two years. Defoliation had a more pronounced impact than browsing and on some of the analyzed variables. Studies that simultaneously evaluate the combined effects of multiple factors on the performance of NTFPs are essential for addressing management scenarios that reflect the realities faced by various species in the field. Despite being protected under the Mexican Red List (NOM-059-SEMARNAT-2010), this endemic and threatened species is still harvested and managed by various groups who rely on it for their livelihoods. Understanding the effects of harvesting, browsing and other potential factors affecting performance of these palms will be crucial for informing effective management and future conservation efforts.

## ACKNOWLEDGEMENTS

This research was conducted in the Monte Mojino Reserve (ReMM) which is within Área de Protección de Flora y Fauna Sierra de Álamos-Río Cuchujaqui and managed by Naturaleza y Cultura-Sierra Madre A.C. We thank the management, operational and administrative staff for their support in logistics and field work, especially Miguel Ángel Ayala Mata and Adriana Álvarez. We thank Sr. Jesús Álvarez (Don Chuy) for the facilities to carry out the experiment at the Rancho Los Llanos.

### Funding

This study was supported by the postdoctoral program of the Institute for Conservation Research- San Diego Zoo Global (2011-2014), the Lakeside Foundation and Coordinación de la Investigación Cientifica- Universidad Michoacana de San Nicolás de Hidalgo. Franceli Macedo-Santana was supported by ''Programa Becas Nacionales CONACyT'' with master's degree grant (Grant No. 352452) and ''Becas Mixtas CONACyT'' (290842) (now SECIHTI) to conduct an academic visit to the University of Hawai'i at Manoa. The funders had no role in study design, data collection and analysis, decision to publish, or preparation of the manuscript.

### Grant Disclosures

The following grant information was disclosed by the authors:
Institute for Conservation Research- San Diego Zoo Global (2011-2014).
Lakeside Foundation.
Coordinación de la Investigación Cientifica- Universidad Michoacana de San Nicolás de Hidalgo.
Programa Becas Nacionales CONACyT: 352452.
''Becas Mixtas CONACyT'': 290842.

### Competing Interests

The authors declare there are no competing interests.

Christa M. Horn is employed by San Diego Zoo Wildlife Alliance.

### Author Contributions

- Franceli Macedo-Santana conceived and designed the experiments, performed the experiments, analyzed the data, prepared figures and/or tables, authored or reviewed drafts of the article, and approved the final draft.
- Christa Horn conceived and designed the experiments, performed the experiments, authored or reviewed drafts of the article, and approved the final draft.
- Tamara Ticktin analyzed the data, authored or reviewed drafts of the article, and approved the final draft.
- María Teresa Pulido Silva analyzed the data, authored or reviewed drafts of the article, and approved the final draft.
- Bryan A. Endress conceived and designed the experiments, performed the experiments, analyzed the data, prepared figures and/or tables, authored or reviewed drafts of the article, and approved the final draft.
- Leonel Lopez-Toledo conceived and designed the experiments, performed the experiments, analyzed the data, prepared figures and/or tables, authored or reviewed drafts of the article, and approved the final draft.

### Field Study Permissions

The following information was supplied relating to field study approvals (i.e., approving body and any reference numbers):

Permit to collect leaves from Dirección General de Vida Silvestre-Secretaría de Medio Ambiente, Recursos Naturales (No. SGPA/DGVS/01991/10 and No. SGPA/DGVS/00837/14).

## Data Availability

The data is available in the Supplementary File.

## Supplemental Information

Supplemental information for this article can be found online at http://dx.doi.org/10.7717/peerj.19266#supplemental-information.

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
