# Peer review of "Individual and demographic responses of the palm Brahea aculeata to browsing and leaf harvesting in a tropical dry forest of Northwestern Mexico"

_PeerJ, doi:10.7717/peerj.19266_

## Round 0.1 · original submission · Minor Revisions

We have received four reviews for your manuscript and all reviewers concurred that it was an interesting and potentially important study for the field. They have provided minor comments that will help further improve the manuscript. In particular, reviewer 1 and reviewer 2 provided specific suggestions regarding aspects of the methods, results and discussion that could be clarified and/or more detailed.

Reviewer 1 ·

Basic reporting

The study developed in this manuscript is original as it deals with an important issue for the sustainable management of the endemic palm Brahea aculeata. The article is well written in general and the language is scientifically appropriate. However, few errors persist in the manuscript that the authors may found in my comments below. The background clearly situates the study and give the justification with relevant literature cited in the text. Figures are for good quality and the tables captions is understandable. The article is well structure and we could find any important information for the understanding of the paper.

Experimental design

The methods employed by the authors were appropriate, scientifically sound to address these important issues of the resilience of palm to multiple factors such as leaf harvesting and browsing. Indeed, they measure browsing and harvesting pressure in addition to individual and population traits that may respond to these factors over three consecutive years. They follow up more than thousands individual in this study which appropriate make they conclusion strong and relevant at scientific level. However, to make the method reproducible by others, the authors should indicate how and when they measure (which instruments and the period of each data collection ) the data at the individual level, this includes the length of the leave and the stem.

Validity of the findings

The finding outlines here indicate that Brahea aculeata is resilient at least for the two years to leaves harvesting and browsing. This finding is supported by a robust data (more than thousands of individuals from juveniles and adults were considered ) collected through three consecutive years from 2011 to 2014 by the authors. I think the outcomes of the study is valid at the scientific level.

Additional comments

This is a point-by-point comments I am doing to make the article clearer to readers and correct some mistakes in the writing.
Comment 1: Line 111-113: Some studies reported no effect of harvesting on individual palm’s vital rates (growth, reproduction and survival) when the harvest is restricted to a few defoliation events, even with complete defoliation. It is necessary to give the reference of these studies (at least one ) in the end of the sentence. That may situate the reader to go and have a look for further detail information if needed.
Comment 2: Line 150 “…as well as the and the interaction of these two factors…” the author should remove the “and” to make the sentence more understandable.
Comment 3: Line 184: what do the authors mean by intensive management? Is it overexploitation? Please make it clearer!

Comment 4: In the study species description section, it is important to give a picture of some part of the species to make it recognizable to all reader as Brahea aculeata is endemic to Northern Mexico. Information such as the subfamily to which the species belong and the ecology of the species is also important to be reported here.
Comment 5: In the result section, the authors affirmed that “Specifically, for the case of mortality, browsing and 296 harvesting did not show any effect”. (Line 296), however, they explained the methods that they do not analyze the mortality due to low case of dead individuals recorded at the end of the experiment (Lines 255-256). This seems confusing, if the data show that browsing and harvesting did not affect the case of mortality, this mean that the data was analyze, and the author should correct this in the method section and give the appropriate statistic that sustain it. if not, which will be the best way as they already stated in the method section, the affirmation in the result section maybe not appropriate here.
Comment 6: Line 311: “higher individual” instead of “larger individual”. Correct it throughout the manuscript
Comment 7: Line 418: the authors stated that “There are other species such as Chamaedorea, which had been reported to be more susceptible to leaf harvesting”. The authors should “spp.” after the genus name to specify that it is many species within the genus they are talking about.
Comment 8: In the tables, please use “p-value” instead of “p valor”!

Reviewer 2 ·

Basic reporting

The manuscript entitled “Individual and demographic responses of the palm Brahea aculeata to browsing and leaf harvesting in a tropical dry forest of Northwestern Mexico” is an important case of NTFPs. The article is written in professional, unambiguous English, with clear articulation of the research goals and findings.
The introduction provides a robust background on the ecological and economic importance of palms as non-timber forest products (NTFPs). A few sentences in the introduction and discussion are overly dense and could benefit from simplification (e.g., lines 70-76 and 385-390).

Experimental design

The choice of three years as the study duration is reasonable; however, the manuscript would benefit from a brief justification of why this period is sufficient to observe cumulative effects (e.g., lines 380-385).

Validity of the findings

The interpretation of results is consistent with the research objectives, providing a nuanced understanding of Brahea aculeata's resilience to disturbances. However, the low mortality observed raises questions about long-term demographic trends. While the authors acknowledge this limitation, they could expand on how future studies might address it (lines 390-405).
The manuscript claims the species is resilient to defoliation (lines 390-392), but the drop in traits in year three suggests thresholds might have been approached. Clarify if this is a temporary decline or indicative of long-term effects.
Consider discussing potential genetic or physiological adaptations that might explain the observed resilience, linking this to broader conservation strategies.
Enhance the discussion on how findings can be generalized to other NTFP species or similar ecosystems, potentially broadening the impact of the study.
Incorporate a clearer distinction between short-term resilience and long-term sustainability in the management recommendations.

Additional comments

Suggestions
Please cite the following relevant studies in the handicrafts section, you are not enforced to cite them, but these are relevant and recent.
ABDULLAH, A., KHAN, S. M., AHMAD, S., ZEB, S. A., HAQ, Z. U., & BALSLEV, H. (2024). On the Trail of the Mazri Palm (Nannorrhops ritchieana) in Pakistan. Palms, 68, 26.
Abdullah, Khan, S. M., Pieroni, A., Haq, Z. U., & Ahmad, Z. (2020). Mazri (Nannorrhops ritchiana (Griff) Aitch.): a remarkable source of manufacturing traditional handicrafts, goods and utensils in Pakistan. Journal of ethnobiology and ethnomedicine, 16, 1-13.
Khan, S. M., Haq, Z. U., Khalid, N., Ahmad, Z., & Ejaz, U. (2022). Utilization of three indigenous plant species as an alternative to plastic can reduce pollution and bring sustainability to the environment. In Natural Resources Conservation and Advances for Sustainability (pp. 533-544). Elsevier.

·

Basic reporting

It is an interesting article and meets the standards of the journal

Experimental design

Experimental design is good and has been explained well

Validity of the findings

Findings are clear and valid

Additional comments

I recommend acceptance with minor grammatical corrections.

Reviewer 4 ·

Basic reporting

Dear Author,
I read your invaluable paper about Individual and demographic responses of the palm Brahea aculeata to browsing and leaf harvesting in a tropical dry forest of Northwestern Mexico with an interesting conclusion about the positive effect of browsing on the studied species as an indirect effect of livestock activity.
1. Include a Latin name in parenthesis.
2. Similarity in structure should be preserved, stems, bark, fruits, and leaves
3. Do not cite more than 3 references for sentences.
4. I think it is better to use NTFPs-providing species instead of NTFPs.
5. Explain about the number of plots and plot size.
6. How did you define the intensity of harvest for low and high harvest?
7. Edit non-timber forest products (NTFP) to NTFPs
8. Comment 3 for the discussion section too, please.
9. Our results have important implications for the management of the species, as to date B. aculeata is a red-listed species under Mexican and international norms, and no permissions are legally allowed. How we can have a management plan for a species at the red list? It is prohibited to harvest legally.
10. Investing for a red-list species is possible?

Experimental design

NA

Validity of the findings

NA

Annotated reviews are not available for download in order to protect the identity of reviewers who chose to remain anonymous.

---

## Round 0.2 · accepted · Accept

I am satisfied with the final revisions made to the manuscript.